# Assessment of the Chemical State of Bottom Sediments in the Eutrophied Dam Reservoir in Poland

**DOI:** 10.3390/ijerph17103424

**Published:** 2020-05-14

**Authors:** Aleksandra Ziemińska-Stolarska, Ewa Imbierowicz, Marcin Jaskulski, Aleksander Szmidt

**Affiliations:** 1Faculty of Process and Environmental Engineering, Lodz University of Technology, 90924 Lodz, Poland; ewa.imbierowicz@p.lodz.pl; 2Faculty of Geographical Sciences, University of Lodz, 90139 Lodz, Poland; marcin.jaskulski@geo.uni.lodz.pl (M.J.); aleksander.szmidt@geo.uni.lodz.pl (A.S.)

**Keywords:** dam reservoir, bottom sediments, eutrophication, spatial distribution, GIS

## Abstract

The aim of the presented research was to examine the concentration of biogenic compounds and heavy metals in the bottom sediments of the Sulejów Reservoir (Central Poland) from October 2018. Based on the obtained research results, maps of the spatial distribution were prepared. The following parameters were analyzed: total phosphorus (TP), total Kjeldahl nitrogen, total organic carbon (TOC), ratio of total organic carbon to nitrogen (C:N), organic matter content as well as Cd, Cr and Pb concentrations. The sediments were collected at 28 sampling sites, covering the whole area of the reservoir. The differences in the content of individual biogenic compounds result from the composition of the debris applied by the supplying rivers, as well as the content of this elements in the water, long retention time (40 days) and depth from which the tested sediments were taken. The distribution of examined compounds was largely influenced by the agricultural activity in the studied area, as well as the presence of ports and recreational points. Based on the measurements, the highest amounts of biogenic components deposit in sediments of deep parts in slow-flowing waters, in stagnation zones, areas adjacent to arable land, and the sites where fine-size fractions prevail in the deposited material. Biogenic compounds in sediments of the Sulejów Reservoir showed a pattern of gradual increase along the reservoir from lower values in the back-water part. A similar relationship is visible for heavy metals. Referring to the ecotoxicological criteria, it can be stated that bottom sediments from the Sulejów Reservoir collected in 2018 are not toxicologically contaminated in terms of cadmium, lead and chromium content.

## 1. Introduction

Determining the chemical status of bottom sediments and water quality in dam reservoirs is an important aspect in basin management [1,2,3]. The chemical composition of the bottom sediments and the heavy metals content reflects human developments and impacts on the catchment and reservoir’s natural environment [4,5,6]. Sediments accumulating at the bottom of water reservoirs are their integral part and fulfill an important role in controlling the cleanliness of the aquatic ecosystem [7,8].

Dam reservoirs are vulnerable to the eutrophication process caused by biogenic substance loads (among others, nitrogen and phosphorus compounds) delivered to the reservoir both from point and diffuse sources of pollution, such as sewage discharges from urbanized areas or runoff from agricultural areas [9,10]. The nutrients can be deposited in sediments and then released to enter the biogeochemical cycles again [5].

In reservoirs with a high trophic level, with intensive accumulation of organic matter in sediments, oxygen deficits are often observed in the bottom layer of water. When anoxic conditions occur in bottom sediments, various compounds may increase in interstitial waters (through dissolution or reduction) and diffuse from the sediments into the lower water layer [11].

The location and depth of the reservoir has a large impact on both the water quality, as well as the composition and characteristics of the sediments—whether it is located in an ecologically clean (natural) or anthropogenically changed area. Accumulation properties mean that pollutant concentrations in sediments are often many times higher than in water alone [12,13]. This allows detection and observation of changes in the content of undesirable substances even with little water pollution.

Composition of the reservoir sediments is related to many factors, including climate, river basin lithology, soils and land use management. These factors influence the efficiency of chemical and mechanical weathering and erosion processes within the basin. Sediments act as sinks for different type of pollutants, which achieve their higher concentrations than in water and may be chemically transformed and deposited on the bottom or liberated to overlying water under certain conditions and being biologically available. Therefore, they may be potentially dangerous especially in dam reservoirs gathering water for drinking purposes, agricultural irrigation or being recreationally used [14,15].

The nature of monitoring the quality of surface waters and sediments results from the guidelines contained in the Directive 2000/60/EC of the European Parliament and of the Council of 23 October 2000 establishing a framework for Community action in the field of water policy (Water Framework Directive). This document defines the objectives and perspectives regarding information on monitoring of water reservoirs, rivers and bottom sediments [16].

Poland is characterized by high seasonal and spatial variability of water resources. Therefore, in order to increase the efficiency of water management and ensure flood protection, dam reservoirs have been constructed for over 50 years. The largest number of retention reservoirs were constructed in lowland areas of Poland. Their location in the agricultural landscape means that they are exposed to a high supply of biogenic compounds mainly nitrogen and phosphorus [4].

The Sulejów water reservoir is of interest due to the possibilities of using its potential in the development of the region, including recreation and tourism. At present, this form of reservoir utilization is not possible due to the progressing eutrophication phenomenon during the summer season. The area of the reservoir was qualified as a region of special landscape values. Around the reservoir, four nature reserves exist: Lubiaszów, Błogie, Gaik and Twarda. In 1994 the Sulejów Landscape Park, with the area of 17,444 km^2^ was established [17].

All rivers naturally transport sediments. However, when the flow velocity and energy of the water is reduced, most of them settles along the bottom of the reservoir where it becomes trapped, rather than continuing downstream. Over a period of years and without sustainable management, the sediment deposits will gradually displace the volume that was previously used for water storage. As water storage volume is lost, the beneficial uses such as water supply and flood control–also decline and can eventually be lost [18,19]. Among the main silting factors are:Type and amount of debris;Dynamics of water flow through the reservoir;Construction, depth and use of the catchment;The intensity of precipitation and their sum resulting from climatic conditions;Wrong location of drain facilities in the catchment area.

According to the International Commission on Large Dams (ICOLD) assessment, the total capacity of artificial water reservoirs in the world (including also dam reservoirs <15 m high) is about 7 billion m^3^ of which 380 billion m^3^ is the dead volume of these reservoirs. The annual weight of sediments carried by rivers (dragged and lifted debris) is estimated at around 24–30 billion tons with flow volumes of 40,000 km^3^. Taking into account the variable content of sediments at various flows and assuming an average amount of approximately 0.6–0.7 t/1000 m^3^, it is estimated that only in reservoirs created by dams operated for 30–40 years, approximately 1400 million m^3^ of debris is being deposited [20]. Due to the importance of the subject, which is silting up of water reservoirs, monitoring and analysis of the composition of bottom sediments should be an important aspect of managing water reservoirs that are particularly vulnerable to accumulation of sediments.

## 2. Study Area

The Sulejów Reservoir is a shallow lowland reservoir with an average depth of 3.3 m, covering an area of 2700 ha (Table 1). One of the biggest artificial reservoirs in Poland was built by impounding the Pilica River on 138.9 km with a dam, in the years 1969–1973. The Reservoir is being supplied by two tributaries Pilica (83%) and Luciąża (15%)—rivers, which deliver municipal sewage from three cities: Piotrków Trybunalski (73,000 inhabitants), Sulejów (6300 inhabitants), Przedbórz (3600 inhabitants). Apart from the two main supplying rivers, direct catchment constitutes also of nine minor tributaries. Most of them are flowing only intermittently, the biggest are right-hand-side tributaries: Radońka River—length 13 km, supply from Błogie and Zarzęcin Mały—11 km and Struga River—9 km. The percentage of small tributaries, in the direct catchment, in water balance ranges from 0.5% to 2% [17].

The Sulejów Reservoir is located in Central Poland (Figure 1). According to the physico-geographical regionalization conducted by J. Kondracki [21], the Sulejów Reservoir is located in the province of the Polish Uplands (34), the sub-province of the Lesser Poland Upland (342), the macroregion of the Przedbórz Upland (342.1) and the mesoregion of the Sulejów Valley (342.111). The reservoir occupies the northern part of the mesoregion, near the border between the Uplands and the Polish Lowland. Owing to its low elevation (below 200 m ASL), the dominant cover of the neighboring areas with Quaternary sediments and low relief diversity, the study area is treated as a lowland.

Average water retention time of several years is 40 days, which cause that total water exchange in the reservoir takes place 9 times a year. The entire catchment of the discussed reservoir has an agricultural and forest character. Cultivated lands covered 64.21% and forests 30.69% of the total basin area. A similar structure has the arable lands in the vicinity of the reservoir (19.9 km^2^), where the fields pose 60.03% and forests 37.43%, respectively [17].

The Sulejów Reservoir is a ribbon-type reservoir, where two morphologic zones, each influenced by different forcing agents can be distinguished (Figure 2). The first one (consisting of a riverine zone and a transition zone) is the narrow, shallower part of the reservoir, dominated by the river inflows. The second, wide, lacustrine part of the reservoir, is located near the frontal dam. Lacustrine zone is open, the main driving forces mechanism causes the movement of water masses is wind. Main axis runs from southwest to northeast which is close to the direction of winds that ripple and mix the water. As a result, formation of places with stagnant water, on the southern bank of the middle and lower part of the reservoir, is visible.

In the riverine zone the sedimentation of phosphorus associated with mineral and organic suspension and dragging of deposited material on the bottom into the reservoir takes place. Due to high flow rate and turbidity of water in this region, the primary production is relatively low, and consequently introduction of dissolved phosphorus compounds into the biologic cycle is insignificant.

In the transitional zone, with a decrease of water flow rate the phosphorus compounds in fine clay and organic particles are sedimented. Slow-down of water flow rate, decrease of its turbidity and high content of nutrient salts dissolved in water favor an increased primary production. As a result, the role of phosphorus cycle increases. Particulate deposit is susceptible to resuspension, due to which periodic internal supply of phosphorus is possible in this zone.

In the lacustrine zone, dominating elements of the phosphorus cycle are the biologic cycle and sedimentation of dead organic particles. Periodically, under favorable physicochemical conditions, the internal supply, in particular the release of dissolved phosphorus compounds from mineral deposits, can be important for the phosphorus cycle [22,23].

## 3. Methodology

The bottom sediment samples were collected in autumn of 2018, from 28 sites. Sampling points were located randomly to cover the whole reservoir bed area.

During sampling, time and geographical coordinates of the sampling location were recorded. The top layer of the sediment was collected from a depth of 0–15 cm using an Ekman sampler (Figure 3).

The collected sediments were transferred to polyethylene containers and transported to the laboratory in a cooler for the analyses. The test sludges were homogenized and dried first at 20 °C and then to constant weight at 105 °C. The prepared sludges were subjected to chemical analysis. Concentrations of total nitrogen, total phosphorus and total organic carbon were determined in the sediment samples.

Organic matter content was estimated as a loss on ignition at 550 °C and calculated as the difference between total and remaining inorganic matter contents.

Total phosphorus was determined spectrophotometrically. Samples for phosphorus determination were previously mineralized in an Ethos Easy microwave mineralizer.

The sediment samples were examined to determine the total nitrogen content (TN) using Kjeldahl method with a TEKATOR microwave oven and the distillation unit KJELTEC. Nitrogen analysis was carried out in accordance with the PN-73/C-04,576/12 standard.

The content of total organic carbon in the examined bottom sediment samples was determined by coulometric method using the Behr apparatus. For both series of analyzed sediment samples, the ratio of organic carbon content to total nitrogen content (Corg:Nog) was calculated, as well as the percentage of total organic carbon (TOC) [25].

In the series of sediment samples taken from the bottom of the Sulejów Reservoir in the autumn of 2018, the content of heavy metals: cadmium, lead and chromium was examined.

The samples were sieved through a 0.2 mm sieve. Bottom sediments were mineralized in a mixture of HNO_3_ and H_2_O_2_ using the Multiwave 3000 microwave system (Anton Paar).

Heavy metal concentrations in the tested samples were determined using atomic absorption spectrometry techniques: with atomization in flame (ContrAA 300, Analytik Jena)—the following elements were determined: Pb, Cd and with electrothermal atomization (SensAA, GBC)—Cr. Sediment samples were analyzed in triplicates, an accuracy of the performed analyses was tested using reference material CRM 16-05.

The obtained results of the measurements were processed in the Golden Software Surfer 14. (Golden Software, Golden, CO, USA) The kriging technique was applied to provide optimal unbiased interpolation estimation for each point. Preparation of ambient layers was processed in ESRI ArcGIS 10.2 (Esri, Redlands, CA, USA) based on Open Street Map data and General Geographic Database (BDO). The final composition of cartographic studies was made in Corel Draw X7 (Corel Corporation, Ottawa, Canada)

The places of collecting bottom sediment samples from the reservoir are illustrated in Figure 4.

## 4. Results

Test results of selected chemical parameters of bottom sediments in the Sulejów Reservoir are presented in Table 2.

On the basis of the obtained results, maps of the spatial distribution were prepared. The maps are presented in Figure 5, Figure 6, Figure 7 and Figure 8.

The content of total phosphorus in sediments taken from the Sulejów Reservoir in 2018 reached values in the range of 0.72–3.57 g/kg dry matter. TP in the sediment of the Sulejów Reservoir was quite unevenly distributed, somewhat following the distribution of organic matter in sediments. The highest values of TP in sediment were recorded above the narrowing part located above the so-called "small lagoon" and along the left bank to the height of the island (Stations 14, 23 and 25), as well as in the area near the dam (Station 3–2.99 g/kg). High concentration of TP (2.70 g/kg dry matter) was also noted at the left bank of the reservoir below the yacht port in Smardzewice (Station 8).

Similar to nitrogen, the lowest values of TP were recorded in back water where the highest flow rate and shallow depth (1.8 m) is noted (Station 26 and 28; 0.78 and 0.72 g/kg, respectively). In the backwater section TP was mostly transported by tributaries from the catchment area while in the lower part it resulted from organic matter (dead phytoplankton blooms) sedimentation.

Compare to the results provided by Smal [26], the concentrations of total phosphorus in the sediments reach higher values than in other lowland reservoirs in Poland. The TP contents in sediments of the Sulejów Reservoir were higher than sated 10 years earlier in this reservoir [11].

Internal loading caused by P released from anoxic sediments often represents the main summer P load to lakes and can have an immense effect on their water quality, especially the eutrophication process.

The TN content in sediments of the Sulejów Reservoir was more differentiated than TP and varied from 1.07 up to 12.31 g/kg dry matter. The highest total nitrogen concentration was recorded near the water port in Bronisławów (Station 16; 12.31 g/kg) whereas, the lowest concentrations were recorded in the riverine zone. TN concentrations in the sediments in the riverine part oscillated between 1.1 and 3.78 g/kg (Stations 28 and 24, respectively). What attracts attention is the zone of relatively lower nitrogen concentration in the sediments deposited at the frontal dam (Station 27; 1.46 g/kg dry matter)–in the distant, southeastern part of the reservoir. The reason for that is undoubtedly the system of water outflow from the reservoir, which rises transport and outflow of sediments from the area of the frontal dam, especially fine size mineral and organic matter fractions. Along with organic matter, nitrogen is removed. Comparing the obtained results with the work of Smal [26], the TN concentration in the Sulejów Reservoir reach similar mean value—5.04 g/kg while mean TN concentration in two lowland reservoirs in Poland is 5.12 g/kg dry matter—Zemborzycki Reservoir and 2.81 g/kg in Brody Iłżeckie Reservoir.

The content of total organic carbon in the sediments taken from the Sulejów Reservoir in 2018 varied greatly throughout the entire reservoir and reached values in the range of 2.33–140.75 g/kg dry matter. The highest concentration of total organic carbon was recorded near the yacht port in Smardzewice (Station 3; 140.75 g/kg). Other areas of high TOC concentrations were around the water marina in Bronisławów and around the island in the upper part of the reservoir—at the left bank (Station 16 and 21; 125.42 and 116.53 g/kg, respectively).

The carbon-to-nitrogen ratio indicates the sedimentary organic matter transformation, mineralization and humification processes. The C:N ratio in the Sulejów Reservoir‘s sediments ranges from 2.47–13.27 and proves variability of the organic carbon and total nitrogen contents. The mean C:N value was equal to 7.6.

The C:N ratio is higher (greater than 20) in macrophytic and cellulose-rich terrestrial organic matter and lower (between 4 and 10) in algae and phytoplankton [27]. Similar results (from 5.36 up to 10.41) were obtained by Small et al. [26] who examined this parameter in two dam reservoirs in Poland (Zemborzycki and Brody Iłżeckie Reservoirs).

The content of organic matter in sediments taken from the Sulejów Reservoir in 2018 reached values in the range of 0.41%–24.36%. Organic matter content in the sediments is influenced by many factors, e.g., allochthonous discharge of organic matter and nutrients, reservoir morphometry, current velocity (which favors creation of lentic and lotic zones), autogenic primary productivity, diagenesis processes and resuspension.

Organic matter content in the sediment of the Sulejów Reservoir was related to the reservoir depth. Lower amounts (1%–2%) were recorded in a shallow part of the reservoir (Stations 26 and 28; 1.95% and 1% respectively) which reflected poor establishment of littoral flora.

Relatively high concentrations of organic matter were recorded around the former water intake station in Bronisławów village and on the opposite bank of the reservoir, where the water flow rate decreases, and the sediment is more easily deposited (Stations 11 and 16; 19.53% and 21.82% respectively). The lacustrine part of the reservoir was the richest in organic matter (mean value—24.36%).

Analysis conducted by Trojanowska and Jezierski [11] in 2013, who measured organic matter content in the sediments of 4 polish reservoirs (Turawa, Sulejów, Włocławski, Siemianówka), indicate that concentration of OM is lower than 20%, which confirms that the concentration of OM in the Sulejów Reservoir reaches relatively high values.

In 2018, the collected sediments were also tested for heavy metals content: cadmium, lead and chromium. The studied metals accumulated in bottom sediments showed significant spatial distribution across the reservoir (Table 3, Figure 9, Figure 10 and Figure 11). Studies conducted by Baran et al. [28] show that the spatial distribution in heavy metals concentrations among reservoirs is associated with different anthropogenic activities. The metals in sediments come from the catchment via fluvial transport, atmospheric deposition and/or municipal wastewater and industrial sewage [4].

Moreover, shape and morphology of the reservoir, reservoir operation and the biochemical processes modify the heavy metals deposition [13,21,29].

The Sulejów Reservoir is surrounded by an agricultural and forest catchment, hence the contamination of its sediments with heavy metals, seems relatively small. Concentrations of the analyzed heavy metals typically fell within the lowest to average amounts detected in other reservoirs located in Poland [28,30]. The highest amounts of heavy metals were accumulated in the transitional and riverine zone of the reservoir.

The Cd concentrations in the sediments taken from the Sulejów Reservoir reached values in the range of 0.22–1.70 mg/kg dry matter.

The highest cadmium content was determined in the sediments taken form from the vicinity of island in the left, upper part of the Sulejów Reservoir (Station 21; 1.7 mg/kg dry matter) and in the sediments from the lower part, at the right bank, at the height of Karolinów village (Station 8; 1.56 mg/kg dry matter).

The content of lead in sediments was quite diverse in the reservoir area and reached values in the range 1.34–38.75 mg/kg dry matter. The highest values of lead were determined in sediments from the right bank of the lacustrine zone at the height of the Karolinów village (Station 8 and 11; 34.55 and 38.75 mg/kg dry matter, respectively). A relatively high content of lead was also found in sediments collected from the vicinity of island in the upper left part of the lake (Station 21; 33.06 mg/kg dry matter). Similar results have been presented in the studies by Tarnawski and Baran [31] and Baran et al. [32] who examined heavy metal concentration in both lowland and Carpathian polish reservoirs.

The content of chromium was, as in the case of lead and cadmium, diverse throughout the entire reservoir and reached values in the range of 1.61–38.59 mg/kg dry matter.

The highest chromium content was determined in sediments from the lower bank of the riverine part of the reservoir (Station 21; 33.55 mg/kg dry matter). Sediments from the upper part of the reservoir were also characterized by high chromium content (Station 3 and 11; 34.06, 38.59 mg/kg dry matter, respectively).

The sediment quality was compared with the criteria established and used by the Chief Inspectorate of Environmental Protection. Up to 2012, there was one legal act in Poland regarding the quality of sediments. This was the Regulation of the Minister of the Environment of 16 April 2002 on the types and concentrations of substances that cause the spoil is contaminated (Journal of Laws No. 55. item 498). Nowadays—from 2013 onwards—the research in the subsidence monitoring of bottom sediments are carried out based only on geochemical and ecotoxicological criteria [33].

Geochemical criteria developed by the Polish Geological Institute relate to the classification of bottom sediment purity based on the content of selected trace elements [27]. As part of the geochemical criteria, for the purposes of monitoring, a classification has been developed that extracts IV degrees of sediment pollution. Table 4 presents the values of the geochemical background and concentration of heavy metals adopted for individual degrees of sediment contamination and the maximum concentrations of tested metals determined in the sediments collected in 2018 from the Sulejów Reservoir.

According to the above classification, bottom sediments from the Sulejów Reservoir can be classified as non-contaminated in terms of chromium content and slightly contaminated in terms of cadmium and lead content.

According to Namieśnik and Rabajczyk [34] the determination of the chemical species of compounds present in aquatic ecosystems, both in solution and in the bottom sediments, carried out in the context of the bioavailability of the various species and their forms of occurrence, can supply data that are essential for the assessment of threats to the environment. The toxicity of elements, as well as the presence or absence of synergism and antagonism with respect to other elements, and hence their positive or negative effects on the functioning of living organisms, depends on the species in which they occur. In discussions on the problems of contamination it is therefore important to differentiate between particular oxidation states and the forms of occurrence of metals, not only at the instant they enter the environment, but also during their migration and transformations in its different compartments.

Ecotoxicological criteria are based on the PEC (Probable Effect Concentration) indicator, whose value sets the limit for the content of an element or chemical compound above which a toxic effect on organisms is noticeable. Table 5 presents the limit values of the PEC indicator and the concentrations of the studied metals determined in sediments taken from the Sulejów Reservoir in 2018.

Referring to the ecotoxicological criteria, it can be stated that bottom sediments from the Sulejów Reservoir collected in 2018 are not toxicologically contaminated in terms of cadmium, lead and chromium content. The concentrations of the tested metals are well below the specified limits. They overlap with the results of other authors who studied heavy metal concentrations in the sediments of Polish dam reservoirs [4,33,34,35,36].

## 5. Discussion

Intensity of the sedimentation processes, nutrient cycle and internal supply of biogenic compounds in the Sulejów Reservoir reservoirs, has significant spatial differentiation, as a function of the environmental factors, such as:The water flow rateThe abundance of food compounds (fertility)The availability of lightThe content of dissolved oxygen in water

The factors take the form of gradients, oriented most often along the main axis of the reservoir, and in regard to certain factors also vertical gradients, which result in directional hydrodynamic and morphologic changes in the water bodies. The interactions among all these factors influence the rate of suspension sedimentation and sediment texture [21,29]. Due to the high complexity of the shorelines and reservoir beds (narrows, bays, islands, etc.) of the Sulejów Reservoir, water flow is disturbed as illustrated in Figure 12.

Research on flow hydrodynamics in the Sulejów Reservoir in 2015 [37] confirms, that when steady flow pattern develops in the basin, a large region of recirculation is formed bellow the outlet of the reservoir. Figure 12 shows the surface velocity field. The length and color of the vectors depict the magnitude of the velocity, as well as the direction of flow. Such circulating arrangements reached the size of even half kilometer and increased the water retention time. Recirculation zones have also a meaning impact on the distribution of phytoplankton cells during the water blooms. Lacustrine area of the reservoir is characterized by significant slowing the flow <0.1 m/s, which creates favorable conditions for the sedimentation rate.

Generally, the highest concentrations of biogenic components deposit in sediments of deep parts of the Sulejów Reservoir: in slow-flowing waters, in stagnation zones, areas adjacent to arable land, and the sites where fine-size fractions prevail in the deposited material.

Interestingly, satellite maps of the blooms on the Sulejów Reservoir (Figure 13) confirm that the highest concentration of nutrients was recorded in the places of intensive blooms during the growing season. Location of cyanobacteria cells (light green color) is clearly visible and coincide with the circulation areas, obtained from the CFD model (Figure 12). Results show that hydrodynamics affect the water body mainly by wind drift and have a direct influence on the phytoplankton arrangement and bottom sediment concentration distribution.

The Sulejów Reservoir was found to have longitudinal (perpendicular to the dam) zonation of the TN and TP distribution in sediments. The lowest nitrogen and phosphorus content was found in the sediments in the left-bank part of the reservoir, medium along the middle part and the highest in the right-bank area. The zone of lowest contents may be related to the parent river current and strong water waving. In turn, the zone of the highest TN concentrations can be ascribed mainly to a lacustrine environment.

A similar conclusion may also be indicated in the case of organic matter content and total organic carbon in the sediments of the Sulejów Reservoir. A general pattern of increasing content of TOC and OM in the vicinity of the dam was visible. According to Gierszewski et al. [12] organic matter is mainly deposited in the deepest part of water bodies because the rate of its decomposition at low temperature and dissolved oxygen content are lower there compared with the shallow part.

Riverine zones of the reservoirs are shallow and have relatively high flow rates; organic matter is transported there by advective currents, but relatively little of it is deposited. Low concentrations of sedimentary organic carbon in the southern part of the reservoir may be ascribed then to the lower depth and higher water dynamics than other areas. Weaker accumulation of organic matter in the shallow water zone and in the zone of higher water dynamics were confirmed by a number of authors [14,20,22,25,27,36].

A significant interconnection between organic matter and TP content is visible in the reservoir. The main pathway of phosphorus delivery to sediments from organic matter was confirmed. However, it is suspected that TP in the backwater section originated from different processes than the TP in the upper and lower courses of the reservoir. In the backwater section TP was mostly transported by tributaries from the catchment area while in the lower part it resulted from organic matter (dead phytoplankton blooms) sedimentation. According to Wagner and Zalewski [38], organic matter and TP delivered to Sulejów Reservoir depends significantly on the discharge.

Bing et al. [39] suggested that local anthropogenic activities increase contamination of heavy metals in the bottom sediments, especially of upper regions. Saleem et al. [40] observed relatively high content of heavy metals at sites which were adjacent to urban and semi-urban areas. The heavy metals may also come from the discharges of untreated urban or industrial wastes and agricultural runoffs. High heavy metals concentration near the dam is a consequence of changes in hydrological conditions in reservoirs during floods and high flow periods. Many authors emphasize the relationship of heavy metals with organic matter content, which is also visible in the case of the Sulejów Reservoir [31,33,38,41,42,43].

## 6. Conclusions

The Sulejów Reservoir is one of the largest lowland dam reservoirs in Poland. Despite the fact that since 2004 it is no longer a source of drinking water, it is still an important place of recreation. It is a shallow water body in which the quality of sediments directly affects the state of its waters, which have been threatened by eutrophication for years. The conducted analyses led us to draw the following conclusions:

(1) Intensified accumulation of biogenic compounds has been observed especially in the vicinity of dam and places of constrictions, mainly in upper and middle course of the reservoir, what is typical for discharge-dependent sedimentation.

(2) The distribution of biogenic compounds in bottom sediments is largely influenced by agricultural activity in the studied area, as well as the presence of ports and beaches. Mean TP and TN concentrations reach values of 1.74 and 5.04 g/kg, respectively.

(3) The C:N ratio in sediments ranges from 5.24–10.19 and means that the main source of organic matter in sediments is degradable phytoplankton.

(4) Organic matter is a key component of sediment but was present at the rather low level of ca. 8% dry matter. Along the length of the reservoir, the OM contents were found to be progressively higher.

(5) The analysis of the TOC spatial distribution has demonstrated a noticeable longitudinal zonation pattern. The highest values were observed in the lacustrine part of the Sulejów Reservoir reaches the mean value of 55.73 g/kg.

(6) The average values of the metals tested were in intervals: cadmium—0.22–1.70 mg/kg dry matter, lead—1.34–38.75 mg/kg and chromium—1.61–38.59 mg/kg dry matter. These values were comparable to the results obtained by the other authors for the lowland reservoirs in previous years. On a similar level, lead and chromium concentrations were observed in the bottom sediments for the reservoirs: Jeżewo, Jutrosin, Rydzyna, Środa and Września. The average results obtained for lead and cadmium were lower than the averages presented for the Sulejów Reservoir.

(7) The concentrations of heavy metals determined in the samples ranged from the natural content of soils (the geochemical background) to sediments weakly or poorly contaminated.

## Figures and Tables

**Figure 1 ijerph-17-03424-f001:**
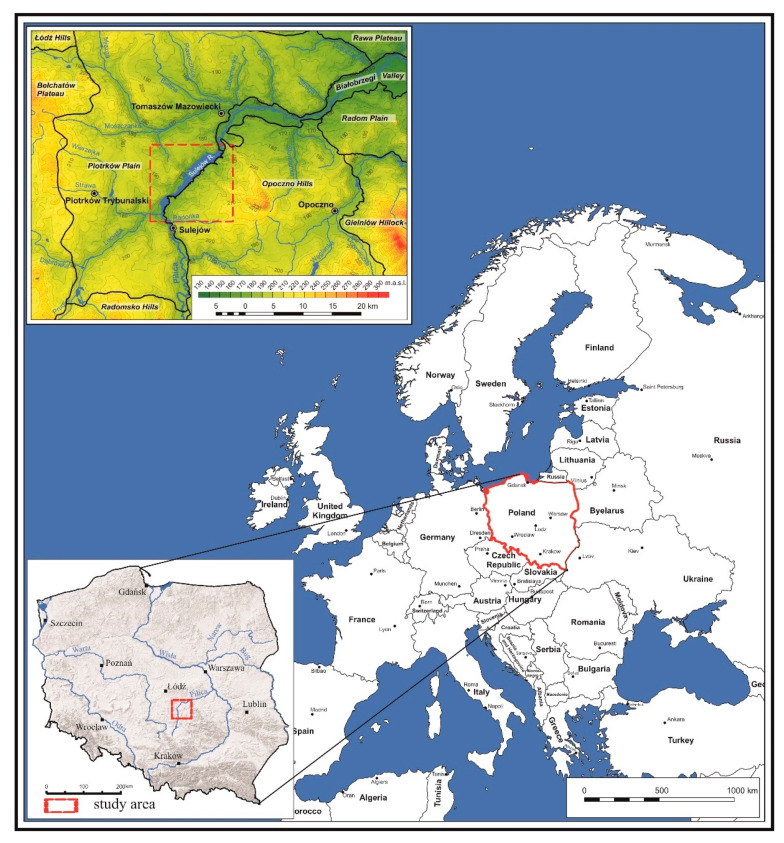
Location of the Sulejów Reservoir.

**Figure 2 ijerph-17-03424-f002:**
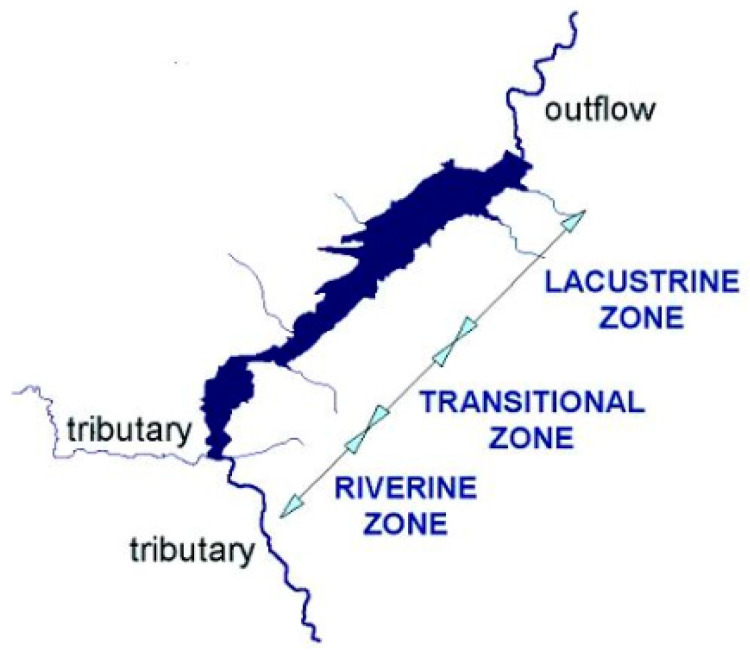
Division of the Sulejów Reservoir into riverine, transitional and lacustrine zone [24].

**Figure 3 ijerph-17-03424-f003:**
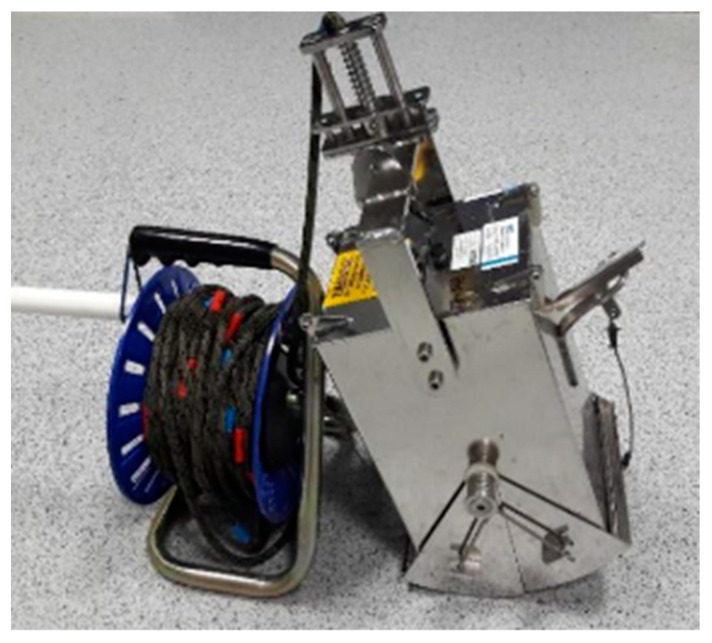
Ekman bottom sediment sampler.

**Figure 4 ijerph-17-03424-f004:**
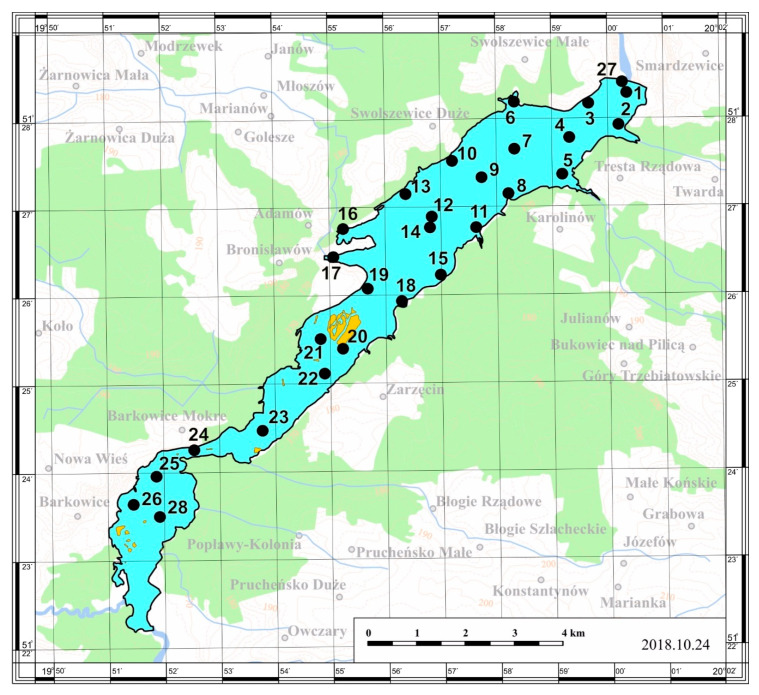
Sampling point locations in the Sulejów Dam Reservoir.

**Figure 5 ijerph-17-03424-f005:**
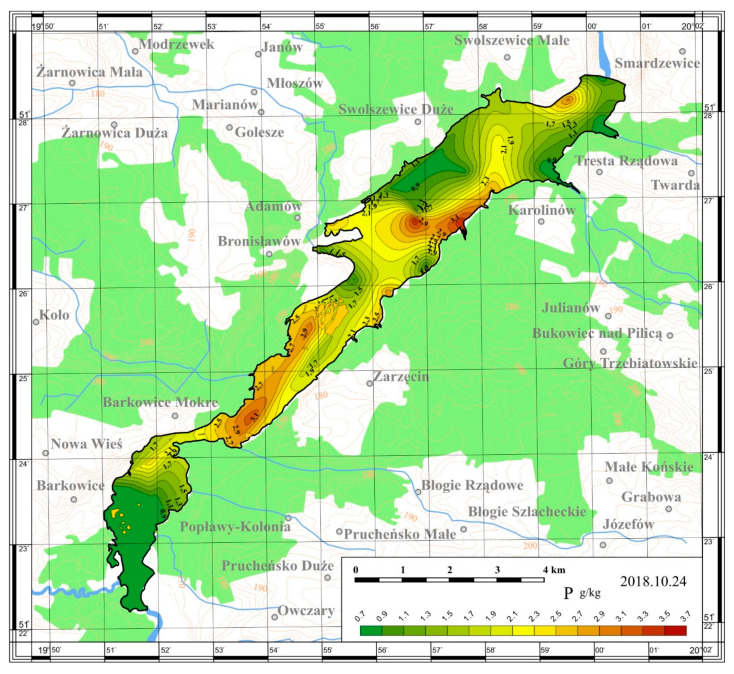
Spatial distribution of the total phosphorus content in the bottom sediments of the Sulejów Reservoir.

**Figure 6 ijerph-17-03424-f006:**
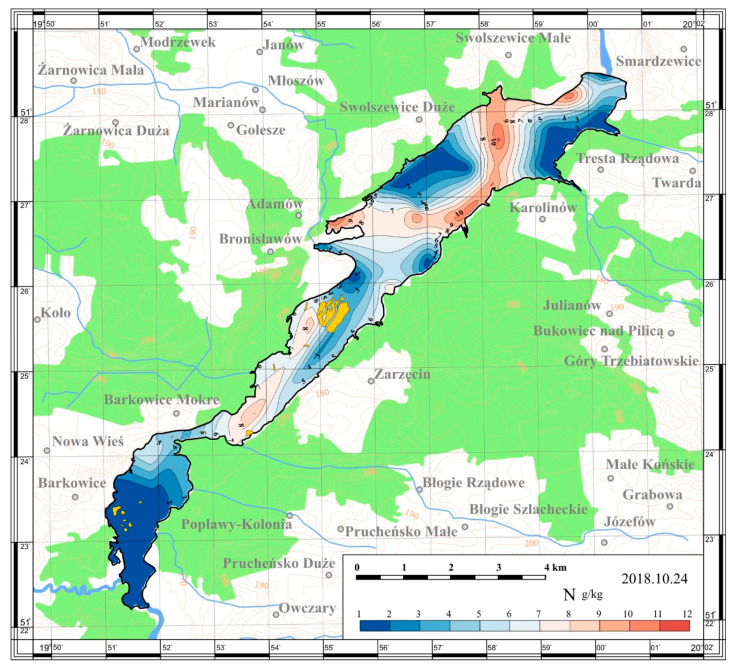
Spatial distribution of the total nitrogen content in the bottom sediments of the Sulejów Reservoir.

**Figure 7 ijerph-17-03424-f007:**
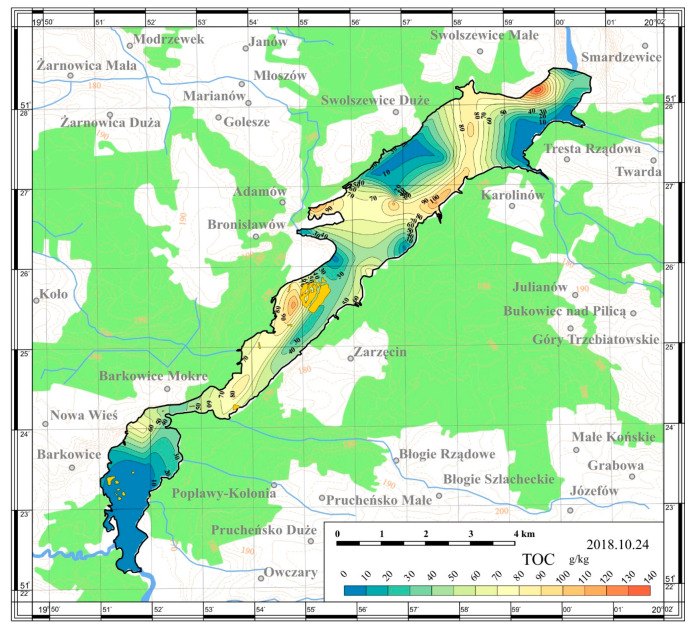
Spatial distribution of the total organic carbon content in the bottom sediments of the Sulejów Reservoir.

**Figure 8 ijerph-17-03424-f008:**
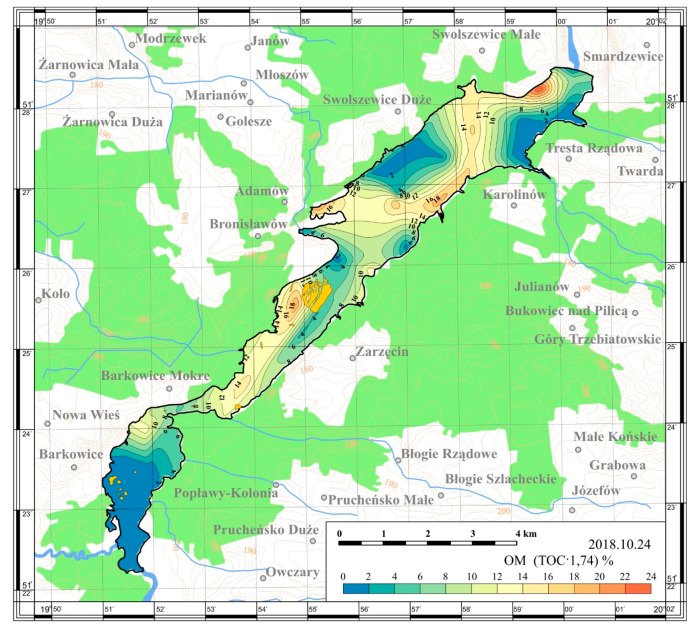
Spatial distribution of organic matter content in bottom sediments of the Sulejów Reservoir.

**Figure 9 ijerph-17-03424-f009:**
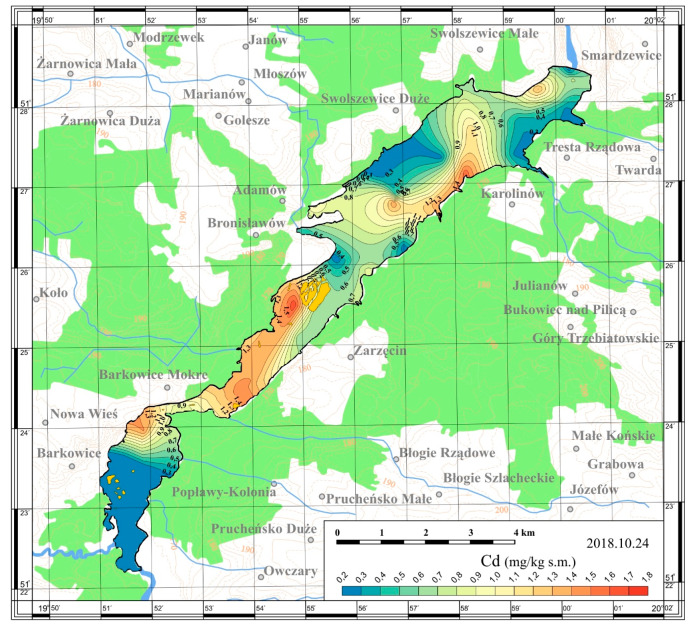
Spatial distribution of cadmium in bottom sediments of the Sulejów Reservoir.

**Figure 10 ijerph-17-03424-f010:**
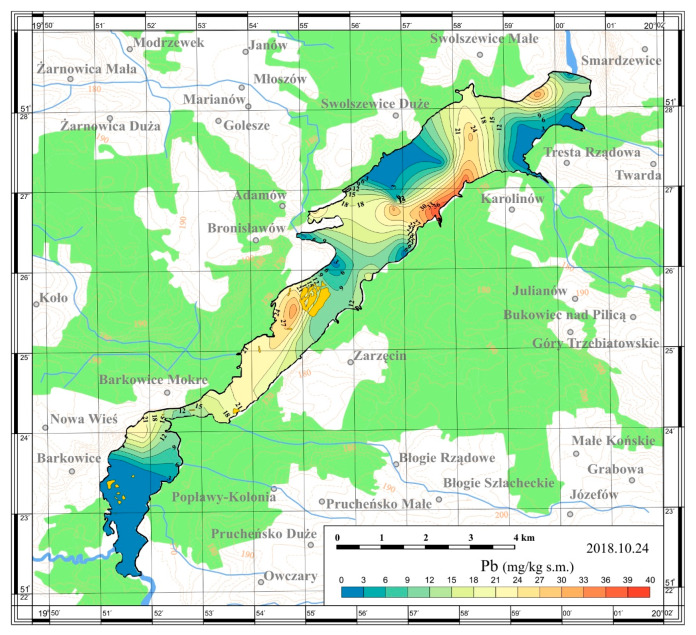
Spatial distribution of lead in bottom sediments of the Sulejów Reservoir.

**Figure 11 ijerph-17-03424-f011:**
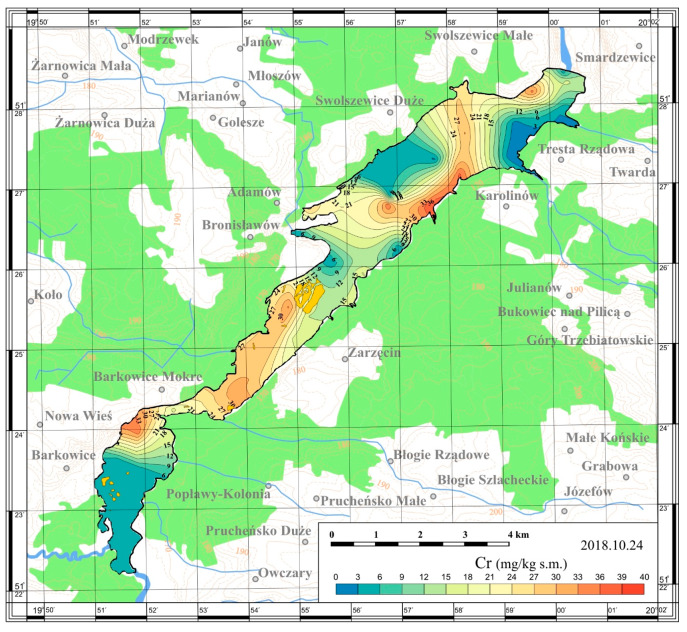
Spatial distribution of chromium in bottom sediments of the Sulejów Reservoir.

**Figure 12 ijerph-17-03424-f012:**
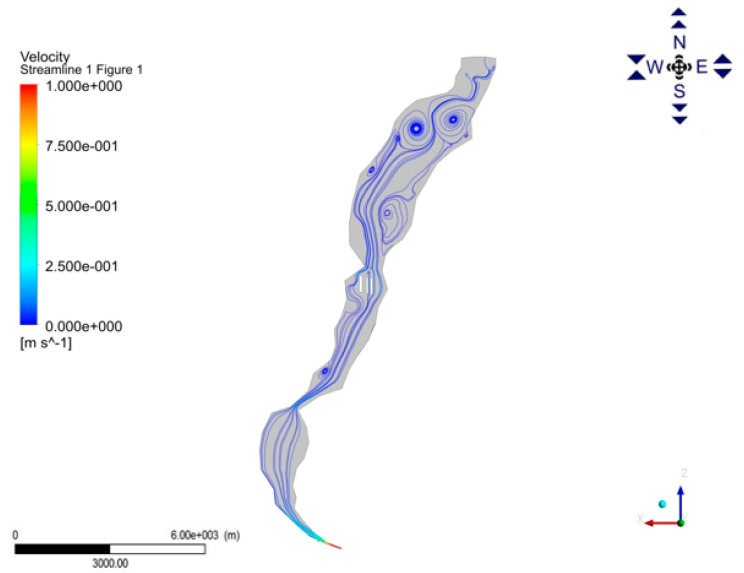
Surface velocity stream lines in the Sulejów Reservoir in October 2015 [2].

**Figure 13 ijerph-17-03424-f013:**
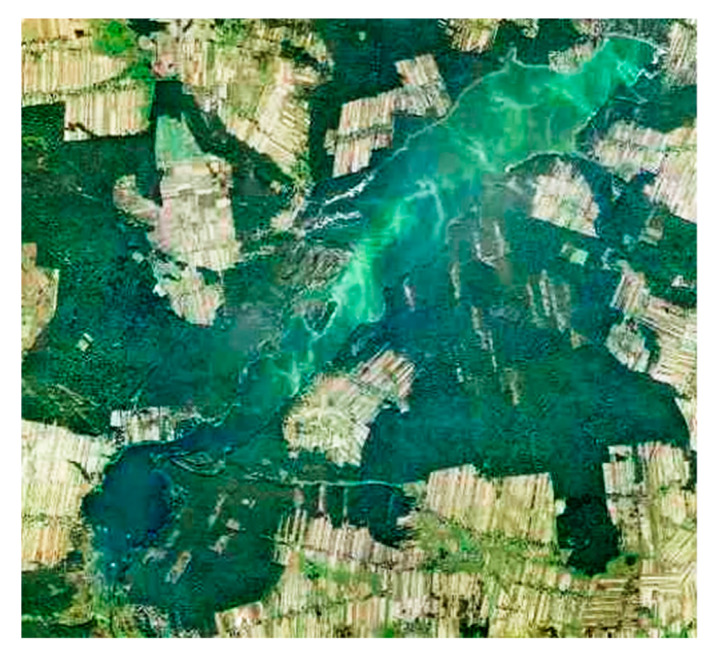
Satellite map of the Sulejów Reservoir (Source: geoportal.gov.pl).

**Table 1 ijerph-17-03424-t001:** Morphometric parameters of the Sulejów Reservoir.

Parameter	Unit	Value
Water surface	ha	2700
Maximum depth	m	11
Average depth	m	3.3
Maximum width	km	2.1
Minimum width	km	1.0
Length	km	17.1
Length of the coastline	km	58
Usable capacity	m ³	61 × 10^6^
Maximum capacity	m³	75 × 10^6^
Catchment area	km²	4900

**Table 2 ijerph-17-03424-t002:** Chemical composition of bottom sediments in the Sulejów Reservoir October 2018.

No	Geographical Location		Depth (m)	g/kg Dry Matter			C:N	Organic Matter = Carbon (%) x 1.72
	Width	Length		Kjeldahl Nitrogen	TOC	Total Phosphorus		
1	51.471722	20.005750	9.3	5.60	40.98	1.26	7.32	7.13
2	51.465694	20.003277	3.5	1.19	2.93	0.80	2.47	0.51
3	51.469777	19.994361	10.0	10.96	140.75	2.99	12.84	24.36
4	51.463333	19.988583	6.5	1.34	2.33	1.38	1.74	0.41
5	51.456388	19.986361	2.0	1.07	3.98	0.74	3.73	0.69
6	51.470166	19.972250	3.7	9.45	87.01	1.50	9.21	15.14
7	51.461250	19.972194	9.0	11.24	96.34	2.30	8.57	16.76
8	51.452833	19.970277	10.0	10.13	96.66	2.70	9.54	16.82
9	51.455916	19.962333	5.2	1.25	9.53	0.83	7.65	1.66
10	51.459083	19.953611	3.0	1.17	4.37	0.79	3.73	0.76
11	51.446472	19.960555	6.7	10.93	112.26	3.39	10.27	19.53
12	51.448583	19.947388	5.5	3.64	18.05	1.00	4.96	3.14
13	51.452888	19.939638	2.2	1.28	7.80	0.85	6.07	1.36
14	51.446527	19.946805	7.8	10.14	104.07	3.57	10.27	18.11
15	51.437555	19.949861	3.0	1.15	7.68	0.82	6.66	1.34
16	51.446416	19.920944	4.5	12.31	125.42	2.71	10.19	21.82
17	51.441111	19.917944	5.0	2.01	13.14	0.95	6.53	2.29
18	51.432638	19.938194	4.9	8.00	76.4	2.85	9.55	13.29
19	51.435055	19.928055	3.4	1.17	5.18	0.88	4.42	0.90
20	51.423722	19.920500	4.5	2.81	20.56	1.66	7.31	3.58
21	51.425611	19.913888	3.9	8.78	116.53	3.10	13.27	20.28
22	51.419055	19.915000	4.1	2.79	19.85	1.52	7.11	3.45
23	51.408361	19.896305	3.5	9.09	88.55	3.28	9.75	15.41
24	51.404861	19.875916	3.0	3.78	31.78	1.97	8.41	5.53
25	51.399916	19.864611	2.7	5.90	75.41	2.49	12.79	13.12
26	51.394638	19.857722	2.0	1.47	11.22	0.78	7.64	1.95
27	51.473777	20.004472	8.5	1.46	8.26	0.76	5.66	1.44
28	51.392277	19.865472	1.8	1.10	5.76	0.72	5.24	1.00

**Table 3 ijerph-17-03424-t003:** Concentration of trace elements in bottom sediments.

No.	Geographical Location		Depth (m)	mg/kg Dry Matter		
	Width	Length		Cd	Pb	Cr
**1**	51.47172222	20.00575000	9.3	0.73	14.91	16.00
**2**	51.46569444	20.00327778	3.5	0.28	2.59	3.16
**3**	51.46977778	19.99436111	10.0	1.32	32.48	34.06
**4**	51.46333333	19.98858333	6.5	0.31	1.68	1.61
**5**	51.45638889	19.98636111	2.0	0.28	1.88	1.61
**6**	51.47016667	19.97225000	3.7	0.66	18.18	26.11
**7**	51.46125000	19.97219444	9.0	1.20	28.19	30.35
**8**	51.45283333	19.97027778	10.0	1.56	34.55	36.52
**9**	51.45591667	19.96233333	5.2	0.25	1.97	2.44
**10**	51.45908333	19.95361111	3.0	0.26	1.85	4.51
**11**	51.44647222	19.96055556	6.7	1.34	38.75	38.59
**12**	51.44858333	19.94738889	5.5	0.43	3.58	4.71
**13**	51.45288889	19.93963889	2.2	0.22	1.34	2.99
**14**	51.44652778	19.94680556	7.8	1.41	31.31	35.89
**15**	51.43755556	19.94986111	3.0	0.24	1.68	2.84
**16**	51.44641667	19.92094444	4.5	1.06	26.72	29.22
**17**	51.44111111	19.91794444	5.0	0.45	2.63	4.16
**18**	51.43263889	19.93819444	4.9	0.87	16.78	18.25
**19**	51.43505556	19.92805556	3.4	0.22	1.44	3.03
**20**	51.42372222	19.92050000	4.5	0.63	9.25	14.95
**21**	51.42561111	19.91388889	3.9	1.70	33.06	33.55
**22**	51.41905556	19.91500000	4.1	0.67	14.58	17.98
**23**	51.40836111	19.89630556	3.5	1.49	22.47	32.27
**24**	51.40486111	19.87591667	3.0	0.80	10.3	17.57
**25**	51.39991667	19.86461111	2.7	1.41	22.45	34.27
**26**	51.39463889	19.85772222	2.0	0.24	2.47	5.71
**27**	51.47377778	20.00447222	8.5	0.24	1.95	4.07
**28**	51.39227778	19.86547222	1.8	0.23	1.90	4.39

**Table 4 ijerph-17-03424-t004:** Comparison of maximum concentrations of tested heavy metals determined in sediments from the Sulejów Reservoir with the values adopted in the sediment classification bottom based on geochemical criteria [33].

Metal	mg/kg Dry Matter					
	Geochemical Background	Maximum Determined Metal Concentration	Class I Uncontaminated	Class II Slightly Polluted	Class III Contaminated	Class IV Heavily Contaminated
Chromium	6	38.59	50	100	400	>400
Cadmium	<0.5	1.70	0.7	3.5	6	>6
Lead	15	38.75	30	100	200	>200

**Table 5 ijerph-17-03424-t005:** Comparison of maximum concentrations of selected heavy metals determined in sediments from the Sulejów Reservoir with the values adopted in the classification of bottom sediments on the basis of ecotoxicological criteria [33].

Metal	mg/kg Dry Matter	
	PEC Limit Value	Metal Concentration
Chromium	111	38.59
Cadmium	4.98	1.70
Lead	128	38.75

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
