# Peer review of "Assessment of the Chemical State of Bottom Sediments in the Eutrophied Dam Reservoir in Poland"

_ijerph, 2020, doi:10.3390/ijerph17103424_

Round 1

Reviewer 1 Report

This work requires many improvements. I suggest that the authors check this work again.
Please pay attention to what should be in the following chapters
Summary - it should concern a small introduction, the research area, what was examined and what results came out
Methodology - if something is discussed, done, everything must be included in the methodology
Results - here we just discuss our results
Discussion - here we describe your results compared to the literature
Summary - here should be a summary of your research

comments in the remarks

Author Response

Dear Reviewer:

 Thank you for your comments and suggestions concerning our manuscript entitled “Assessment of the Chemical State of Bottom Sediments in the Eutrophied Dam Reservoir in Poland”. Those comments are valuable and very helpful for revising and improving our paper. We have studied all comments carefully and have made correction point by point. Revised parts are marked in yellow in the paper. All manuscript chapters have been revised according to the guidelines, both the abstract and the introduction methodology, results, discussion and conclusions

Reviewer1

Q1: Abstract was revised and include also obtained results.

Q2,Q3,Q4: The text in the Introduction pagaraf has been changed.

Q5: Additional sentence was added to the text form line 101 to 103.

Q6: Table 1 was added to the text. Line 106.

Q7:Map presenting the location of the reservoir has been changed.

Q8: Literature has been competed.

Q9: Link to the table was add to the main text.

Q10: Information concerning the standard was added. Lines 172-174.

Q11: Name of the parameter was changed.

Q12: Numeration of figures have been changed

Q13: This part was removed from the text.

Q14:This part was added to the discussion. Lines 367-377.

Q15: Additional references were added.

Q16: The letter size has been changed

Q17: This part was changed. Lines 322-327

Q18: Reference was added

Q19: Conclusions have been changed

Q20: Summary has been changed.

Once again, thank you very much for your comments and suggestions.

With regards Aleksandra Ziemińska-Stolarska

Email: aleksandra.zieminska-stolarska@p.lodz.pl

Reviewer 2 Report

The authors studied the pollution status of a water reservoir in Poland. The material has the potential of being published. However, in the current state the quality of the writing of the paper is not suitable even for peer-review. The introduction is unclear and gives no hypothesis or course of the following study. Materials and methods lack the proper description of the methodology. The results are chaotic and unclear.

The abstract does not reflect the content of the paper. There are multiple technical issues like the use of commas instead of points as decimal separator, data rounding, etc.

What are the geographical coordinates? Are those longitude and latitude? 

The author should rewrite their paper (probably consulting a colleague with a good skill of academic writing) and then it can be reconsidered again.

Currently, it is impossible to evaluate the scientific quality and novelty since nothing is clear.

Author Response

Dear Reviewer:

Thank you for your comments and suggestions concerning our manuscript entitled “Assessment of the Chemical State of Bottom Sediments in the Eutrophied Dam Reservoir in Poland”. Those comments are valuable and very helpful for revising and improving our paper. We have studied all comments carefully and have made correction point by point.

Revised parts are marked in yellow in the paper. All manuscript chapters have been revised.

Q1:The authors studied the pollution status of a water reservoir in Poland. The material has the potential of being published. However, in the current state the quality of the writing of the paper is not suitable even for peer-review.

The paper has been improved, all sections have been completed.

The introduction is unclear and gives no hypothesis or course of the following study. Materials and methods lack the proper description of the methodology. The results are chaotic and unclear.

The introduction has been changed, method descriptions, results , as well as conclusions and summary. The revised article also contains new literature items that complete the discussion.

The abstract does not reflect the content of the paper.

Abstract has been changed.

There are multiple technical issues like the use of commas instead of points as decimal separator, data rounding, etc.

All technical aspects were improved.

What are the geographical coordinates? Are those longitude and latitude? 

Geographical coordinates were added.

The author should rewrite their paper (probably consulting a colleague with a good skill of academic writing) and then it can be reconsidered again.

Currently, it is impossible to evaluate the scientific quality and novelty since nothing is clear.

Once again, thank you very much for your comments and suggestions.

With regards Aleksandra Ziemińska-Stolarska

Email: aleksandra.zieminska-stolarska@p.lodz.pl

Reviewer 3 Report

This is a well-written manuscript. It centres on the chemical composition and heavy metal content of the bottom sediments in the dam reservoir located in the central Poland.

The main problem of this paper is the lack of novelty or at least the lack of efforts to replace this study in a broader context and to explain in what way this paper can be useful to a large audience.

The structure and ordering of the information in the various sections needs to be carefully reviewed. The structure should follow logically as background, methods, results then discussion/findings and keep only the relevant information in these sections.

Lines 293-294. Are there more information about human activity and temperature measurements are needed to confirm this conclusion?

Technical comments:
The contours of iso-values have been used to illustrate the distribution of different elements. I propose select another way of illustration like graduated and proportional symbol maps.

I would recommend this paper for publication after MINOR revisions.

Author Response

Dear Reviewer:

Thank you for your comments and suggestions concerning our manuscript entitled “Assessment of the Chemical State of Bottom Sediments in the Eutrophied Dam Reservoir in Poland”. Those comments are valuable and very helpful for revising and improving our paper. We have studied all comments carefully and have made correction point by point.

Revised parts are marked in yellow in the paper.

This is a well-written manuscript. It centres on the chemical composition and heavy metal content of the bottom sediments in the dam reservoir located in the central Poland.

The main problem of this paper is the lack of novelty or at least the lack of efforts to replace this study in a broader context and to explain in what way this paper can be useful to a large audience.

The information in the article has been supplemented with a comparison of results with other lowland dam reservoirs in Poland.

The structure and ordering of the information in the various sections needs to be carefully reviewed.

All the chapters in the manuscrict has been carrefouly  reviewed and improved.

The structure should follow logically as background, methods, results then discussion/findings and keep only the relevant information in these sections.

The structure of the manuscript has been updated.

Lines 293-294. Are there more information about human activity and temperature measurements are needed to confirm this conclusion?

The text was suplemented by additional conclusions.

Technical comments:

The contours of iso-values have been used to illustrate the distribution of different elements. I propose select another way of illustration like graduated and proportional symbol maps.

The figures have been changed.

I would recommend this paper for publication after MINOR revisions.

Once again, thank you very much for your comments and suggestions.

With regards Aleksandra Ziemińska-Stolarska

Email: aleksandra.zieminska-stolarska@p.lodz.pl

Reviewer 4 Report

Manuscript Number: ijerph-782453

Title: Assessment of the Chemical State of Bottom Sediments in the Eutrophied Dam Reservoir in  Poland

Reviewer' comments:

The paper ijerph-782453 aims to examine the amount of biogenic compounds and heavy 62 metals contained in the bottom sediments of the Sulejów Reservoir from April and October 2018.

The main objective of this study was to analyze the concentration of: total phosphorus, total Kjeldahl nitrogen, total organic carbon, ratio of total organic carbon to nitrogen (C: N) and organic matter as well as Cd, Cr, Pb concentrations, assess contamination and analyze the spatial variability of chemical pollutions and identify potential sources and factors determining the concentration and spatial distribution.

The references are relevant, correctly referenced and internationally evaluated. Keys studies included are appropriate. The aim of the paper is clear. The findings of the paper and how they did it are clear. The title is informative and relevant. The paper has very interesting results.

The introduction states the topic of the research, explain clearly the main idea. The research question is outline and justified what is already known about the topic.

The reviewer wants to know if the following paragraph is from a single reference or not:

P2, L78 to L85: “Sedimentation and loss of significant active volumes of water reservoirs is a serious problem in global water management. According to the ICOLD assessment, the total capacity of artificial water reservoirs in the world (including also dam reservoirs <15 m high) is about 7 billion m3 of which 3 80 billion m3 is the dead volume of these reservoirs. The annual weight of sediments carried by rivers (dragged and lifted debris) is estimated at around 24-30 billion tonnes with flow volumes of 40,000 82 km3. Taking into account the variable content of sediments at various flows and assuming an average amount of approx. 0.6¸0.7 t/1000 m3, it is estimated that only in reservoirs created by dams operated for 30¸ 40 years, approx. 1,400 million m3 of debris is being deposited [22].”

If yes, please make the following change:

According to El-Radaideh [22] "Sedimentation and loss of significant active volumes of water reservoirs is a serious problem in global water management. According to the ICOLD assessment, the total capacity of artificial water reservoirs in the world (including also dam reservoirs <15 m high) is about 7 billion m3 of which 3 80 billion m3 is the dead volume of these reservoirs. The annual weight of sediments carried by rivers (dragged and lifted debris) is estimated at around 24-30 billion tonnes with flow volumes of 40,000 82 km3. Taking into account the variable content of sediments at various flows and assuming an average amount of approx. 0.6¸0.7 t/1000 m3, it is estimated that only in reservoirs created by dams operated for 30¸ 40 years, approx. 1,400 million m3 of debris is being deposited" [22].

Is ICOLD the acronym for International Commission on Large Dams. If yes, specify it in the text. If not, clarify this term in the text.

Results and discussion section

The authors made very good discussions on the quantitative information obtained. However, the reviewer notes the insufficiency of comparative analyzes between the results obtained in this study and those available (for the same parameters) in the scientific literature available on the characterization of the chemical state of bottom sediments in dam reservoir. Therefore, he asks the authors to carry out this comparative analysis if possible for each of the parameters studied.

P13, L272-L274: ”According to the above classification, bottom sediments from the Sulejów Reservoir can be classified as non-contaminated in terms of chromium content and slightly contaminated in terms of cadmium and lead content.”

The reviewer wants this interpretation to be deepened. Indeed, it is based on the idea that each of these metals act in isolation without taking into account the possible interactions that they can have between them, and that they can have with the ligands. The reviewer wants this interpretation to be deepened. Indeed, it is based on the idea that each of these metals act in isolation without taking into account the possible interactions that they can have between them, and that they can have with the ligands. According to NamieÅ›nik& Rabajczyk (2010) “The determination of the chemical species of compounds present in aquatic ecosystems, both in solution and in the bottom sediments, carried out in the context of the bioavailability of the various species and their forms of occurrence, can supply data that are essential for the assessment of threats to the environment. The toxicity of elements, as well as the presence or absence of synergism and antagonism with respect to other elements, and hence their positive or negative effects on the functioning of living organisms, depends on the species in which they occur. In discussions on the problems of contamination it is therefore important to differentiate between particular oxidation states and the forms of occurrence of metals, not only at the instant they enter the environment, but also during their migration and transformations in its different compartments.”

In the light of new information which will be brought to the results and discussion, the reviewer asks the authors to revise the conclusions of the paper.

Overall statement

The paper needs minor revisions before being accepted for publication in ijerph.

Reviewer’ References

NamieÅ›nik, J., & Rabajczyk, A. (2010). The speciation and physico-chemical forms of metals in surface waters and sediments. Chemical Speciation & Bioavailability, 22(1), 1-24.

Author Response

Dear Reviewer:

Thank you for your comments and suggestions concerning our manuscript entitled “Assessment of the Chemical State of Bottom Sediments in the Eutrophied Dam Reservoir in Poland”.

Those comments are valuable and very helpful for revising and improving our paper.

We have studied all comments carefully and have made correction point by point.

 Revised parts are marked in yellow in the paper.

The paper ijerph-782453 aims to examine the amount of biogenic compounds and heavy 62 metals contained in the bottom sediments of the Sulejów Reservoir from April and October 2018.

The main objective of this study was to analyze the concentration of: total phosphorus, total Kjeldahl nitrogen, total organic carbon, ratio of total organic carbon to nitrogen (C: N) and organic matter as well as Cd, Cr, Pb concentrations, assess contamination and analyze the spatial variability of chemical pollutions and identify potential sources and factors determining the concentration and spatial distribution.

The references are relevant, correctly referenced and internationally evaluated. Keys studies included are appropriate. The aim of the paper is clear. The findings of the paper and how they did it are clear. The title is informative and relevant. The paper has very interesting results.

The introduction states the topic of the research, explain clearly the main idea. The research question is outline and justified what is already known about the topic.

The reviewer wants to know if the following paragraph is from a single reference or not:

P2, L78 to L85: “Sedimentation and loss of significant active volumes of water reservoirs is a serious problem in global water management. According to the ICOLD assessment, the total capacity of artificial water reservoirs in the world (including also dam reservoirs <15 m high) is about 7 billion m3 of which 3 80 billion m3 is the dead volume of these reservoirs. The annual weight of sediments carried by rivers (dragged and lifted debris) is estimated at around 24-30 billion tonnes with flow volumes of 40,000 82 km3. Taking into account the variable content of sediments at various flows and assuming an average amount of approx. 0.6¸0.7 t/1000 m3, it is estimated that only in reservoirs created by dams operated for 30¸ 40 years, approx. 1,400 million m3 of debris is being deposited [22].”

If yes, please make the following change:

According to El-Radaideh [22] "Sedimentation and loss of significant active volumes of water reservoirs is a serious problem in global water management. According to the ICOLD assessment, the total capacity of artificial water reservoirs in the world (including also dam reservoirs <15 m high) is about 7 billion m3 of which 3 80 billion m3 is the dead volume of these reservoirs. The annual weight of sediments carried by rivers (dragged and lifted debris) is estimated at around 24-30 billion tonnes with flow volumes of 40,000 82 km3. Taking into account the variable content of sediments at various flows and assuming an average amount of approx. 0.6¸0.7 t/1000 m3, it is estimated that only in reservoirs created by dams operated for 30¸ 40 years, approx. 1,400 million m3 of debris is being deposited" [22].

Is ICOLD the acronym for International Commission on Large Dams. If yes, specify it in the text. If not, clarify this term in the text.

The text was change according to the suggestion.

The abbreviation was explain in the text.

Results and discussion section

The authors made very good discussions on the quantitative information obtained. However, the reviewer notes the insufficiency of comparative analyzes between the results obtained in this study and those available (for the same parameters) in the scientific literature available on the characterization of the chemical state of bottom sediments in dam reservoir. Therefore, he asks the authors to carry out this comparative analysis if possible for each of the parameters studied.

P13, L272-L274: ”According to the above classification, bottom sediments from the Sulejów Reservoir can be classified as non-contaminated in terms of chromium content and slightly contaminated in terms of cadmium and lead content.”

The reviewer wants this interpretation to be deepened. Indeed, it is based on the idea that each of these metals act in isolation without taking into account the possible interactions that they can have between them, and that they can have with the ligands. The reviewer wants this interpretation to be deepened. Indeed, it is based on the idea that each of these metals act in isolation without taking into account the possible interactions that they can have between them, and that they can have with the ligands. According to NamieÅ›nik& Rabajczyk (2010) “The determination of the chemical species of compounds present in aquatic ecosystems, both in solution and in the bottom sediments, carried out in the context of the bioavailability of the various species and their forms of occurrence, can supply data that are essential for the assessment of threats to the environment. The toxicity of elements, as well as the presence or absence of synergism and antagonism with respect to other elements, and hence their positive or negative effects on the functioning of living organisms, depends on the species in which they occur. In discussions on the problems of contamination it is therefore important to differentiate between particular oxidation states and the forms of occurrence of metals, not only at the instant they enter the environment, but also during their migration and transformations in its different compartments.”

In the light of new information which will be brought to the results and discussion, the reviewer asks the authors to revise the conclusions of the paper.

Thank You very much for the valuable information that was aded to the test, moreover conclusions were changed.

Overall statement

The paper needs minor revisions before being accepted for publication in ijerph.

Once again, thank you very much for your comments and suggestions.

With regards Aleksandra Ziemińska-Stolarska

Email: aleksandra.zieminska-stolarska@p.lodz.pl

Round 2

Reviewer 1 Report

Definitely better, a few suggested amendments :)

Author Response

Dear Reviewer,

Thank you for your suggestions concerning our manuscript entitled “Assessment of the Chemical State of Bottom Sediments in the Eutrophied Dam Reservoir in Poland”.

All your comments were taken into account and the paper was revised according to your suggestions.

Q1: In the introduction part, the paragraphs were adjusted.

Q2: Lines 195-198 have been moved to the Methodology chapter

Q3: Table 3, the graphical location in the table was repated to indicate that heavy metal concentrations were measured at the same locations as the nutrient measurements, deleting location information could be confusing.

With regards Aleksandra Ziemińska-Stolarska

Reviewer 2 Report

The paper was not improved in the revision, except for the discussion and abstract. The only thing that was done, the already extensive length was further overgrown. Consider making supplementary information to the paper and putting all less important results there.

In the introduction, give a brief background and state the hypothesis and/or the aim of your study rather than giving extensive unclear generalization. It is clear for the reader that aquatic systems should be under strict pollution control. Line 92 - who is El-Radaideh so the authors expect the readers to give him/her so much credit? 

Methods: The procedure for metal determination is not explained so the quality of the results cannot be evaluated. Please give the information on the instruments, chemicals and most importantly reference materials used for validation.

Lines 117-118 What are the numbers in brackets?

Results: Lines 189-194 Please remove this

Lines 194-198 This belongs to the Methods part, section Data processing, or something like that.

Again! In English comma is not a decimal separator. It is the point so not 10,0 but 10.0; check and correct that everywhere

Author Response

The paper was not improved in the revision, except for the discussion and abstract. The only thing that was done, the already extensive length was further overgrown. Consider making supplementary information to the paper and putting all less important results there.

In the introduction, give a brief background and state the hypothesis and/or the aim of your study rather than giving extensive unclear generalization. It is clear for the reader that aquatic systems should be under strict pollution control. Line 92 - who is El-Radaideh so the authors expect the readers to give him/her so much credit? 

Line 92 has been removed as suggested by the reviewer.

Methods: The procedure for metal determination is not explained so the quality of the results cannot be evaluated. Please give the information on the instruments, chemicals and most importantly reference materials used for validation.

Additional information has been aded to the text. L179-185

Lines 117-118 What are the numbers in brackets?

This is the decimal code for physical-geographical units for the International Division of the World into physical-geographical regions.

According to Solon (2018)

Results: Lines 189-194 Please remove this

Lines 189-194 were removed from the text

Lines 194-198 This belongs to the Methods part, section Data processing, or something like that.

Lines 194-198 were moved to the Methodology section

Again! In English comma is not a decimal separator. It is the point so not 10,0 but 10.0; check and correct that everywhere

All separators were correct in the whole text